# Memory persistence enhancement by post-learning moderate exercise requires *de novo* protein synthesis in the dorsal hippocampus

**Koshiro Inoue** [1*], **Masahiro Okamoto**[2,3], **Takemune Fukuie**[4], **Hideaki Soya**[2,3,5], **Akihiko Yamaguchi**[1]

**1** School of Rehabilitation Sciences, Health Sciences University of Hokkaido, Ishikari-Tobetsu, Hokkaido, Japan, **2** Laboratory of Exercise Biochemistry and Neuroendocrinology, Institute of Health and Sport Sciences, University of Tsukuba, Ibaraki, Japan, **3** Division of Sport Neuroscience, Kokoro Division, Advanced Research Initiative for Human High Performance (ARIHHP), Institute of Health and Sport Sciences, University of Tsukuba, Ibaraki, Japan, **4** School of Nursing and Social Services, Health Sciences University of Hokkaido, Ishikari-Tobetsu, Hokkaido, Japan, **5** Center for Cybernics Research, University of Tsukuba, Ibaraki, Japan

* ikoshiro@hoku-iryo-u.ac.jp

## Abstract

Acute moderate-intensity exercise (AME) after learning has been reported to exogenously boost consolidation of hippocampus-dependent memory, resulting in improved long-term persistence. However, the neuronal mechanism remains poorly understood. Short-term, hippocampus-dependent memory produced by weak encoding can be transformed into long-term memory through an immediate, strong behavioral event, which causes overlapping activation of the hippocampus. Hippocampal *de novo* protein synthesis is essential for achieving memory consolidation in this way. As AME activates the hippocampus, enhanced memory consolidation through post-learning AME may also be mediated by protein synthesis in the hippocampus. To test this hypothesis, this study first attempted to establish a rat model for enhancing memory consolidation via post-learning AME with the object location (OL) test, a hippocampus-dependent spatial memory task. This study used adult male Sprague-Dawley rats, and the AME load was based on the running speed corresponding to the rats' lactate threshold (20 m/min) for 20 min. We then examined the effects of the protein synthesis inhibitor anisomycin (ANI), injected into the dorsal hippocampus, on AME-induced OL memory consolidation. In the OL test, the OL memory encoded with 5 min of learning was retained for at least 1 hr but was lost after 24 hr. With a single bout of AME immediately after the 5 min of OL learning, the memory persisted for 24 hr, indicating AME-induced memory consolidation. The AME-induced OL memory consolidation did not occur when ANI was injected into the dorsal hippocampus immediately or 4 hr after OL learning. These findings support the hypothesis that post-learning AME-induced memory consolidation depends on new-protein synthesis in the dorsal hippocampus and highlight the value of AME after

**Data availability statement:** All relevant data are within the paper and its Supporting Information files.

**Funding:** This study was supported by grants from Japan Society for the Promotion of Science (26750307, 23K10637), and a grant from Advanced Research Initiative for Human High Performance (ARIHHP), University of Tsukuba. The funders had no role in study design, data collection and analysis, decision to publish, or preparation of the manuscript.

**Competing interests:** The authors have declared that no competing interests exist.

learning as a strategy for enhancing memory consolidation. This is a potential base model for future research examining the mechanism behind boosting memory consolidation with exercise.

## Introduction

While a single bout of exercise can improve episodic memory function [1–3], the timing of such exercise interventions in relation to the memory formation process (i.e., encoding, consolidation, and retrieval) critically influences acute exercise and memory interactions [4,5]. A meta-analysis that evaluated the timing-dependent effects of exercise intervention on memory capacity has reported that acute exercise performed before or after, but not during, memory encoding enhances episodic memory function, and, thus, prolongs memory persistence [6]. Notably, the memory boosting effects are greater when acute exercise is performed after, rather than before, memory encoding, that is during the period of memory consolidation. However, the neuronal mechanisms underlying post-encoding exercise-enhanced memory consolidation remain poorly understood.

The fact that the persistence of episodic memory is exogenously enhanced by exercise after learning aligns well with the framework of the behavioral tagging (BT) hypothesis [7]. Memory is thought to be stored in two sequentially linked but temporally distinct forms: short-term memories (STMs), which last minutes to hours and long-term memories (LTMs), which can persist for days, weeks, or even longer [8–10]. The BT hypothesis is that STMs produced by weak encoding during the learning task can be transform into LTMs when an unrelated, strong behavioral event occurs shortly before or after the weak encoding. It is noteworthy that the BT process cannot be achieved unless the brain areas activated by the learning task, which produce only weak encoding, and those activated by the strong behavioral event overlap [11]. Even at the cellular level, the formation of an overlapping memory engram during these two separate events has been shown to underlie the BT process [12]. Moreover, the transformation of STMs into LTMs caused by a strong behavioral event requires *de novo* protein synthesis [7,13]. The weak encoding sets a behavioral tag that is independent of protein synthesis, while the strong event is thought to promote protein synthesis, which, in turn, generates plasticity-related products (PRPs) [11]. Once a PRP is captured by the behavioral tag, the functions and structures of the activated synapses are potentiated over the long term, thereby forming the basis for LTMs [14]. In animal studies, the BT process and its mechanisms have been widely reported using hippocampus-dependent behavioral tasks [13].

The object location (OL) test, a hippocampus-dependent spatial memory task often used in animal models, leverages rodents' innate tendency to explore objects in novel locations more frequently than those in familiar locations [15]. Previous studies have found that the hippocampus, especially the dorsal area, is critical for OL memory formation. For instance, lesions of the hippocampus [16–18] or partial disruptions of activity and plasticity in its dorsal area [19–21] impair OL memory formation. A

past study examining time-dependent changes in OL memory after encoding has shown that rodents can discriminate the object found in a novel-location when the retention interval between the learning and test phases is 1–2 h, but not when this interval is over 4 hr [22]. This retention-interval time-dependent elimination of OL memories also depends on the duration of learning: OL memories disappear within 24 hr after 5–10 min of learning, but persist for 24 hr after 20 min of learning [23]. On the other hand, administration of brain-derived neurotrophic factor (BDNF), a possible product of protein synthesis, into the dorsal hippocampus after only 5 min of learning results in retention of an OL memory 24 hr later [24].

It has been reported that post-learning exercise increases memory persistence in declarative and behavioral memory tasks, which are associated with the hippocampus. For example, a single bout of exercise after associative memory encoding enhances retention performance and improves the consistency of hippocampal activation in response to correct answers in humans [25,26]. At that time, circulating BDNF levels correlate with both retention score and the hippocampal activity [26]. Similar increases in memory persistence induced by post-learning exercise have been reported in rats using the object recognition (OR) test [27–31], but not with the OL test. Although the OR test primarily depends on the perirhinal cortex [32,33], the rat studies have reported that the hippocampus is involved in mediating the exercise-induced enhancement of OR memory persistence [29–31]. Notably, in these studies applied with humans and with rats, the post-learning exercise intervention was conducted at or above moderate intensity. Moderate-intensity exercise activates the dorsal hippocampus and increases the hippocampal expression of BDNF [34–36]. These studies suggest that post-learning, acute, moderate-intensity exercise (AME) enhances memory consolidation in the hippocampus-dependent OL test through protein synthesis in the dorsal hippocampus, resulting in greater OL memory persistence.

To test this hypothesis, we conducted four experiments (Expt.). Expt. 1 had two purposes. First, we confirmed the time-dependent loss of OL memory in rats, that is, whether or not rats could discriminate the displaced object from the non-displaced object after 5 min of learning for the OL test with a short (1 hr) retention interval and with a long (24 hr) retention interval (Expt. 1a). Second, we verified the running speed that corresponds to moderate-intensity exercise by identifying the lactate threshold (LT), a physiological marker of moderate-intensity exercise, during an incremental treadmill test (Expt. 1b). Subsequently, in Expt. 2, we examined whether AME immediately after the OL learning phase would ameliorate the rats' discrimination performance for a retention interval of 24 hr. Finally, we investigated whether or not the administration of the protein synthesis inhibitor anisomycin (ANI) into the dorsal hippocampus immediately after (Expt. 3) or 4 hours after (Expt. 4) the learning phase would eliminate the AME-induced increase in discrimination performance of the OL test with the 24-hr retention interval. Previous studies have reported that exercise leads to immediate activation of the dorsal hippocampus and its monoaminergic system [34,37,38], as well as delayed induction of neuronal plasticity-related molecules, such as BDNF, 2–6 hours after exercise [34,39,40]. Based on these reports, ANI was administered immediately after learning, which is to say just prior to the exercise intervention (Expt. 3), to fully suppress exercise-dependent protein synthesis, or 4 hours after learning (Expt. 4) to target the delayed protein synthesis induced by exercise.

## Materials and methods

### Subjects

All animal care and experimental procedures were performed in accordance with protocols approved by the Animal Ethics and Research Committee of the Health Sciences University of Hokkaido (approval nos. 034 and 104). Ten-week-old male Sprague-Dawley rats (300–340 g; Japan SLC, Shizuoka, Japan) were used as subjects throughout this study. The rats were housed in polycarbonate steel cages (2 rats per cage) with a 12:12 hr light-dark cycle (lights on: 08:00–20:00; temperature 24 ± 2 °C), and given *ad libitum* access to food and water. After a week of environmental acclimatization, the rats were randomly assigned to one of the four experiments: n = 20, 20, 40 and 30 for Experiments 1–4, respectively. For Expt. 1, 10 rats each were assigned to Expt. 1a and 1b. The sample sizes were based on previous studies employing similar experimental designs [11,22,41,42].

## Treadmill running habituation

The rats were habituated to running on a motorized treadmill (KN-73, Natsume Ltd., Tokyo) in the morning over the course of 10 days for a total of 7 running sessions. The running duration per session was 30 min, and the speed was gradually increased from 5 to 25 m/min with no incline (Table 1). A foot shock grid was placed at the rear of the treadmill, set at the minimum level, and rats were shocked only when they refused to run at the set speed. A few shocks were administered during the running habituation, but not during the actual exercise intervention, to minimize aversive stimuli.

## Surgery

Rats were deeply anesthetized with 1–2% isoflurane (Isoflu, DS Pharma Animal Health, Osaka). For Expt. 1b, a silicone catheter was inserted into the rats' jugular veins (32 mm) and fixed with a silk thread [42,43]. For Expt. 3 and 4, 22-gauge stainless steel guide cannulas were bilaterally implanted into the rats' dorsal hippocampi (coordinates: AP −3.6 mm, ML ±2.5 mm from the bregma, and DV −3.0 mm from the skull surface) [44], and fixed on the skull with dental cement and small screws. After the surgeries, the rats were injected with antibiotics (Mycillin Sol; Meiji Seika, Tokyo), then housed individually in cages for 3 days to recover.

## Lactate threshold (LT) identification

LT during the incremental exercise test was identified in accordance with our previous studies [42,43]. The running speed was increased by 2.5 m/min every 3 min beginning at 5 m/min until the rat was running all-out. Blood lactate concentration was measured from blood sampled via the indwelling catheter every 3 min using an automated blood gas analyzer (ABL80 FLEX, Radiometer, Copenhagen, Denmark). LTs were individually identified as the breakpoint in blood lactate versus running speed relationship using the segmented linear regression with a single breakpoint. The segmented linear regression was performed in GraphPad Prism 8 (GraphPad Software, Inc, California, USA), setting no constraint on parameter values. Fig 1 showed a typical example (#4) of identified LT (Fig 1B). The LT, a marker of moderate-intensity exercise, for SD rats was identified at 21.6 ± 0.9 m/min (running speed) on average, with range of 16.2 to 24.8 m/min (Table 2). Based on these results and our previous works [42,43], AME intervention was implemented at a fixed running speed of 20 m/min in experiments 2–4.

## Anisomycin and its infusion

Anisomycin (ANI; Wako, Osaka, Japan) was dissolved in 0.9% saline (Sal; Otsuka, Tokyo, Japan) by adding 1 mol/L hydrogen chloride, and then the pH was adjusted to 7.4 by adding sodium hydroxide. The final concentration was 20 μg/

**Table 1. Treadmill running habituation protocol.**

| Day | Speed (m/min) × Duration (min) |
| --- | --- |
| 1 | Rest×10 + 5 × 10 + 10 × 10 |
| 2 | 5 × 10 + 10 × 10 + 15 × 10 |
| 3 | Rest |
| 4 | 5 × 5 + 10 × 10 + 15 × 10 + 20 × 5 |
| 5 | 10 × 10 + 15 × 10 + 20 × 10 |
| 6 | 10 × 5 + 15 × 10 + 20 × 10 + 25 × 5 |
| 7 | Rest |
| 8 | 15 × 10 + 20 × 10 + 25 × 10 |
| 9 | 15 × 10 + 20 × 10 + 25 × 10 |
| 10 | Rest |

**(A)**

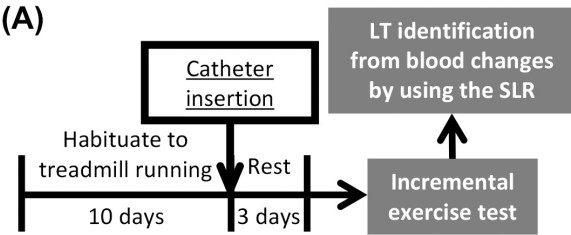

**(B)**

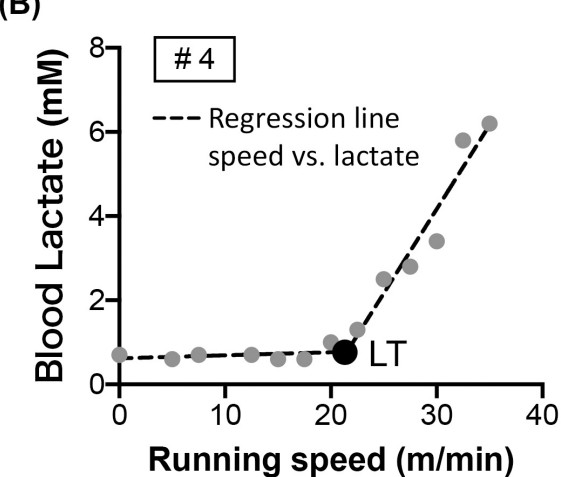

**Fig 1. Determination of lactate threshold (LT) in SD rats.** Expt. 1b: (A) Schematic timeline of the experimental procedures and (B) a typical example (#4) of identified LT.

**Table 2. Individual lactate thresholds (LTs).**

| Subject # | LT point |
|---|---|
| | speed (m/min) |
| 1 | 21.1 |
| 2 | 24.8 |
| 3 | 21.1 |
| 4 | 21.5 |
| 5 | 21.7 |
| 6 | 16.2 |
| 7 | 22.1 |
| 8 | 24.6 |
| Mean ± SE | 21.6 ± 0.9 |

µl. At the time of drug infusion, 26-gauge infusion cannulas were inserted into the guide cannulas until their tips extend 1.0 mm below the guide cannulas. ANI or its vehicle (Sal) were injected into the bilateral dorsal hippocampus through the infusion cannulas which were connected with a PE/PVC tube to a 10 µl Hamilton syringe. The injection rate was driven by a microsyringe pump (ESP-32, EICOM, Kyoto, Japan) at a flow rate of 0.5 µl/min for 2 min (total 1 µl/side). The infusion cannulas were left in place for one additional minute to allow diffusion. At this dose, ANI can inhibit ≧83% of protein

synthesis in the dorsal hippocampus with minimum side effects related to neuronal activity and cell death, and it is effective for approximately 6 hours with a return to base level 9 hours after injection [45–48].

### Object location test

**Apparatus.** An open field arena (81 × 81 × 50 cm) made of gray polyvinyl chloride plastic was used. One of the sidewalls of the arena was designed with a black and white striped pattern as an absolute spatial cue. Before the OL test was performed, rats were handled for 2 min per day and habituated to the test arena without stimulus objects by allowing them to freely explore it for 10 min per day for 5 days. In Expt. 2–4, these procedures were conducted during the last 5 days of treadmill running habituation. The objects employed were glass bottles with a variety of shapes and several duplicates of each object to use interchangeably. All of them were filled with either red or white colored water so that the rats could not move them. A charge-coupled device video camera was suspended above the arena, and the rats' exploratory behavior and total distance moved was automatically monitored and recorded by ANY-maze video tracking software (Stoelting Co, IL, USA) for later analysis.

**OL test procedures.** All OL tests were conducted during the dark period. The light level in the OL testing area was maintained at 6.0–7.5 lux using fluorescent lights. Each OL test trial consisted of a learning phase and a test phase. During the learning phase, rats were allowed to freely explore two identical objects (objects X and Y) placed in the diagonal corners of the test arena 10.0 cm away from the two adjacent walls for 5 min. During the test phase, one of the two familiar objects was placed in the same location as during the learning phase (object F), while the other familiar object was moved to a novel location (object N); both were placed 10.0 cm away from the two adjacent walls. The rats were reintroduced into the arena, and allowed to freely explore the objects for 5 min. After each phase, the floor and walls of the arena and the objects were cleaned with 70% ethanol to avoid potential effects of olfactory cues. A single experimenter assessed the rats' exploration time for each object during the learning and test phases across all experiments by reviewing the video recordings, thereby ensuring consistency in measurement. The experimenter was blind to which animal and treatment condition was being assessed. Exploration was defined as the rats sniffing or touching the objects with their nose and forepaws but not as the rats sitting on or turning around the objects. The exploration time for each respective object and its sum (total object exploration time) during each phase were cumulatively calculated for the entire 5-min duration. To assess location discrimination, a discrimination ratio (DR) was calculated according to the following general formula: (exploration time of Y or N object − exploration time of X or F object)/ total (X + Y or F + N) object exploration time [15]. The relative positions and the role (familiar or novel) of the two identical objects were randomly transposed and counterbalanced for each animal. Each rat underwent two OL test trials, each under different conditions. During the second OL trial, the arena and spatial cues (black-and-white striped wall) remained unchanged from the first trial, while the two identical objects were replaced with two novel identical objects.

In Expt. 1a, the OL test was conducted with a 1-hr or 24-hr interval between the learning and test phases to confirm that the memory formed during the learning phase lasts for an interval of 1 hour but disappears after an interval of 24 hours. To examine the effects of different intervals, all rats underwent each interval in a random order with at least a two-day period between the two OL test trials. A within-subjects design was used for analysis. To assess the reliability of the behavioral data analysis, a second experimenter independently analyzed the same video recordings, and the intraclass correlation coefficient (ICC(2,1)) was calculated between the two experimenters.

In Expt. 2, the effect of the post-learning AME intervention on the OL test performance was examined for the 24-hr interval condition between each phase (learning & test). The 20 min of AME or sedentary control (Sed; placed on a still treadmill) were implemented immediately after the OL learning phase. AME was implemented at a running speed of 20 m/min, which is equivalent to the LT. In order to test and compare the effects of the AME and Sed conditions, all rats underwent the full trial (learning and test phases) for both conditions in random order using a within-subjects design, with more than two days between each trial.

In Expt. 3 and 4, the combined effects of the post-learning AME intervention and the ANI treatment on OL test performance with a 24-hr period were examined to investigate whether AME-enhanced memory consolidation depends on hippocampal protein synthesis. ANI or Sal was injected into the dorsal hippocampus immediately after (Expt. 3) or 4 hours after (Expt. 4) the learning phase. AME and Sed conditions were implemented as in Expt. 2. Similar to Expt. 2, rats underwent the full OL trial (learning and test phases) twice in random order. However, for each trial in Expt. 3 and 4, the AME and Sed conditions were fixed, and the drug interventions (ANI or Sal) were switched. The two trials were separated by more than two days, and the placement of the objects during each phase of each trial was changed randomly.

### Histology

In Expt. 3 and 4, the tip locations of injection cannulas were identified using Nissl staining (Fig 2A). After the 2nd OL test trial, rats were euthanized with a mixture of three of anesthetic agents (0.75 mg/kg of medetomidine, 4.0 mg/kg of midazolam and 5.0 mg/kg of butorphanol), and intracardially perfused with 0.9% saline followed by 4% paraformaldehyde in 0.1 mol/L phosphate buffer. Brains were removed, fixed overnight in paraformaldehyde and then immersed in 30% sucrose at 4°C. They were frozen with carbon dioxide and sectioned in the coronal plane (50 μm) using a cryostat (CM1860, Leica, Heidelberg, Germany). Sections were mounted and air-dried. The dried sections were stained with toluidine blue, dehydrated in ethanol, delipidated in xylene, and cover-slipped with Mount-Quick (Daido Sangyo, Tokyo, Japan). The stained

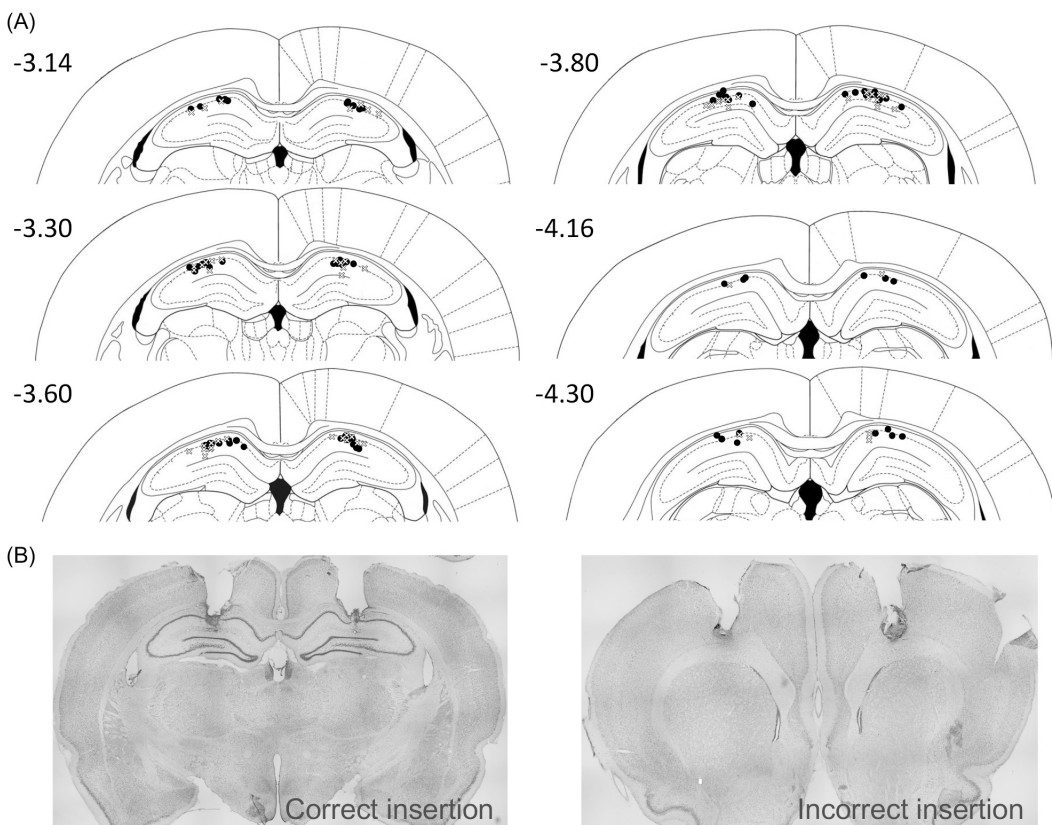

**Fig 2. Locations of the tips of injection cannulas.** (A) Black dots (Expt. 3) and white crosses (Expt. 4) indicate the tips of injection cannulas implanted into the dorsal hippocampus in each animal. Numbers in each brain section represent anteroposterior distance (mm) from bregma [44]. (B) A photomicrograph shows a typical example of correct (left photo) and incorrect (right photo) insertion of cannulas.

sections were visualized with an optical microscope to assess the locations of the tips of injection cannulas (DM6 B, Leica, Heidelberg, Germany).

## Statistical analysis

In this study, nine rats were excluded from the statistical analyses: five exhibiting almost no activity during the OL test (one in Expt. 1a and two each in Expt. 2 & 3), two with blocked catheters (Expt. 1b), one with cannulas not inserted into the target position in the hippocampus (Expt. 3, Fig 2B), and one exhibiting poor running during the AME intervention (Expt. 4). Data are expressed as mean±SE. The normality of the data and homogeneity of variance were assessed using a Shapiro-Wilk test and a Levene's test, respectively. For parametric datasets, paired t-test, two-way repeated measures (RM) ANOVA (Expt. 1a: hour×object; Expt. 2: exercise×object), two-way mixed ANOVA (Expt. 3 & 4: exercise×drug, exercise×trial), and a three-way mixed ANOVA (Expt. 3 & 4: exercise×drug×object) were performed to compare differences between groups or conditions. The Bonferroni *post hoc* test was applied for multiple comparisons in the two-way or three-way RM/mixed ANOVA. DRs were also compared to the theoretical chance level (0.0) with one-sample t-tests in accordance with previous studies [24,49]. The assumption of normality or homogeneity of variance was violated for the following datasets: respective object exploration time during test phase in Expt. 1a, DRs during the learning phase and total/respective object exploration time during the test phase in Expt. 3 and total distance moved during the learning phase and respective object exploration during the test phase in Expt. 4. For these datasets, nonparametric tests were applied, such as Wilcoxon signed-rank test, and two-way or three-way ANOVA with aligned rank transformation (ART) procedure [50]. The difference was considered statistically significant at $p < .05$. Data analyses were performed using SPSS Statistics version 29 (IBM Corp., Armonk, N.Y., USA) and Graphpad Prism 8 (GraphPad Software, Inc, CA, USA). The ANOVAs with ART procedure were carried out with R Studio version 2025.05.0+496 ARTool.

## Results

### Effects of 1- and 24-hour retention intervals on object location memory

In Expt. 1a, behavior analysis using video footage of the rats' movements confirmed that, with no outside intervention, rats could discriminate the novel-location object (N) from the familiar-location object (F) during the OL test with a short (1-hr) retention interval but not with a long (24-hr) retention interval. Further, to ensure the reliability of the behavioral data analysis in this study, the video footage was independently reanalyzed by a different experimenter. The two experimenters' analyses produced similar statistical results as follows.

For the learning phase, total distance moved [paired $t(8) = 0.903$, $p = .393$], total object exploration time [experimenter 1: paired $t(8) = 1.606$, $p = .147$; experimenter 2: paired $t(8) = 1.622$, $p = .143$], respective object exploration time [experimenter 1: $F_{hr×ob}(1,8) = 1.298$, $p = .295$; $F_{hr}(1,8) = 2.580$, $p = .147$; $F_{ob}(1,8) = 0.228$, $p = .646$; two-way RM ANOVA; experimenter 2: $F_{hr×ob}(1,8) = 0.935$, $p = .362$; $F_{hr}(1,8) = 2.632$, $p = .143$; $F_{ob}(1,8) = 0.422$, $p = .534$; two-way RM ANOVA], and DRs [experimenter 1: paired $t(8) = 1.120$, $p = .295$; experimenter 2: paired $t(8) = 1.251$, $p = .246$] did not differ between the 1-hr and 24-hr interval conditions in both experimenters (A in Table 3) . Additionally, DRs did not differ from chance level (0.0) in either condition [experimenter 1: 1-hr; one-sample $t(8) = 0.087$, $p = .933$; 24-hr; one-sample $t(8) = 1.083$, $p = .310$; experimenter 2: 1-hr; one-sample $t(8) = 0.075$, $p = .942$; 24-hr; one-sample $t(8) = 1.233$, $p = .253$] (A in Table 3).

For the test phase, there were no differences between the two conditions in total distance moved [paired $t(8) = 0.424$, $p = .683$; Fig 3B], total object exploration time [experimenter 1: paired $t(8) = 0.636$, $p = .543$; Fig 3C; experimenter 2: paired $t(8) = 0.634$, $p = .544$; Fig 3F], and respective object exploration time [experimenter 1: $F_{hr×ob}(1,24) = 2.786$, $p = .108$; $F_{hr}(1,24) = 0.631$, $p = .435$; $F_{ob}(1,24) = 1.378$, $p = .252$; two-way ART ANOVA; Fig 3D; experimenter 2: $F_{hr×ob}(1,24) = 2.311$, $p = .142$; $F_{hr}(1,24) = 0.878$, $p = .358$; $F_{ob}(1,24) = 0.714$, $p = .406$; two-way ART ANOVA; Fig 3G]. In contrast, DRs were significantly above chance level (0.0) under the 1-hr, but not the 24-hr, retention interval condition [experimenter 1: 1-hr; one-sample $t(8) = 2.934$, $p = .019$, Cohen's $d = .977$: 24-hr; one-sample $t(8) = 0.270$, $p = .794$; Fig 3E; experimenter 2:

**Table 3. OL learning phase measurements.**

| Measurement | | Distance moved (m) | Exploration time | | | DR (%) |
|---|---|---|---|---|---|---|
| | | | Total (s) | X object (s) | Y object (s) | |
| A. Expt. 1a (n = 9) | | | | | | |
| Experimenter 1 | 1H | 20.36±1.27 | 50.19±6.25 | 25.58±4.07 | 24.61±3.07 | −0.01±0.08 |
| | 24H | 18.27±2.06 | 33.39±5.75 | 14.92±3.00 | 18.47±3.50 | 0.13±0.12 |
| Experimenter 2 | 1H | | 60.53±6.91 | 30.36±4.23 | 30.17±3.92 | −0.01±0.08 |
| | 24H | | 42.02±6.31 | 18.79±3.32 | 23.23±4.00 | 0.12±0.10 |
| B. Expt. 2 (n = 18) | | | | | | |
| Sed | | 16.50±1.48 | 50.59±4.65 | 25.54±3.06 | 25.05±2.27 | 0.01±0.05 |
| AME | | 16.27±1.49 | 52.48±5.50 | 24.53±3.66 | 27.95±2.99 | 0.11±0.10 |
| C. Expt. 3 (Sed = 19, AME = 18) | | | | | | |
| Sed | Sal | 23.01±1.23 | 54.33±3.61 | 27.61±2.50 | 26.71±1.93 | −0.01±0.05 |
| | ANI | 22.23±1.04 | 53.34±4.55 | 27.43±2.92 | 25.91±2.57 | −0.01±0.06 |
| AME | Sal | 19.87±1.52 | 43.92±3.71 | 22.02±2.87 | 21.90±2.11 | 0.05±0.09 |
| | ANI | 23.22±1.24* | 49.92±3.44 | 23.71±2.05 | 26.22±2.46 | 0.04±0.06 |
| D. Expt. 4 (Sed = 15, AME = 14) | | | | | | |
| Sed | Sal | 22.53±1.76 | 61.60±4.37*,# | 31.49±4.09 | 30.11±2.02 | 0.02±0.08 |
| | ANI | 23.36±1.30 | 52.61±3.61 | 26.97±2.19 | 25.64±2.17 | −0.03±0.05 |
| AME | Sal | 20.99±1.85 | 47.91±3.14 | 20.96±2.20 | 26.95±2.44 | 0.13±0.07 |
| | ANI | 20.84±2.19 | 53.53±6.05 | 26.84±4.04 | 26.68±2.97 | 0.05±0.10 |

1H: 1-hour retention interval condition; 24H: 24-hour retention interval condition; Sed: sedentary control; AME: acute moderate exercise; Sal: saline; ANI: anisomycin. Data are expressed as mean±SE, * $p < .05$ vs AME/Sal by Bonferroni *post-hoc* test (two-way mixed ANOVA, a simple main effect). # $p < .05$ vs Sed/ANI by Bonferroni *post-hoc* test (two-way mixed ANOVA, a simple main effect).

1-hr; one-sample $t(8)$ =2.493, $p = .037$, Cohen's $d = .831$: 24-hr; one-sample $t(8)$ = 0.025, $p = .981$; Fig 3H]. Also, DRs were significantly higher in 1-hr condition than that in 24-hr condition [experimenter 1: paired $t(8)$ = 2.311, $p = .050$, Cohen's $d = .770$; Fig 3E, experimenter 2: paired $t(8)$ = 2.317, $p = .049$, Cohen's $d = .772$; Fig 3H]. In addition, high inter-rater reliability between the two experimenters was confirmed for the data on respective object exploration time [learning phase: ICC(2,1) =.893, test phase: ICC(2,1) =.861] and on DRs [learning phase: ICC(2,1) =.939, test phase: ICC(2,1) =.959], supporting the reliability of the behavioral data.

## Effects of post-learning acute moderate-intensity exercise on OL test

In Expt. 2, we investigated whether AME implemented immediately after learning could ameliorate the discrimination of the displaced object in the test phase of the OL test for the 24-hr retention interval. During the learning phase, similar learning behavior was observed in Sed and AME conditions (B in Table 3). Specifically, there was no difference between the two conditions in total distance moved [paired $t(17)$ = 0.163, $p = .872$], total object exploration time [paired $t(17)$ = 0.362, $p = .722$], respective object exploration time [$F_{ex×ob}(1,17)$ = 0.688, $p = .418$; $F_{ex}(1,17)$ = 0.131, $p = .722$; $F_{ob}(1,17)$ = 0.404, $p = .533$; two-way RM ANOVA], and DRs [paired $t(17)$ = 0.936, $p = .362$]. Furthermore, DRs for each condition were not different from chance level [Sed: one-sample $t(17)$ = 0.198, $p = .845$; AME: one-sample $t(17)$ = 1.147, $p = .267$].

For the test phase, there was no difference between the two conditions in total distance moved [paired $t(17)$ = 1.286, $p = .216$; Fig 4B] and total exploration time [paired $t(17)$ = 0.898, $p = .382$; Fig 4C]. In contrast, respective object exploration time showed a significant main effect of object (N-object > F-object, $p = .026$; Bonferroni *post-hoc* test), but not of exercise intervention, and no significant interaction between the two factors [$F_{ob}(1,17)$ = 5.377, $p = .033$, $\eta^2_p = .240$; $F_{ex}(1,17)$

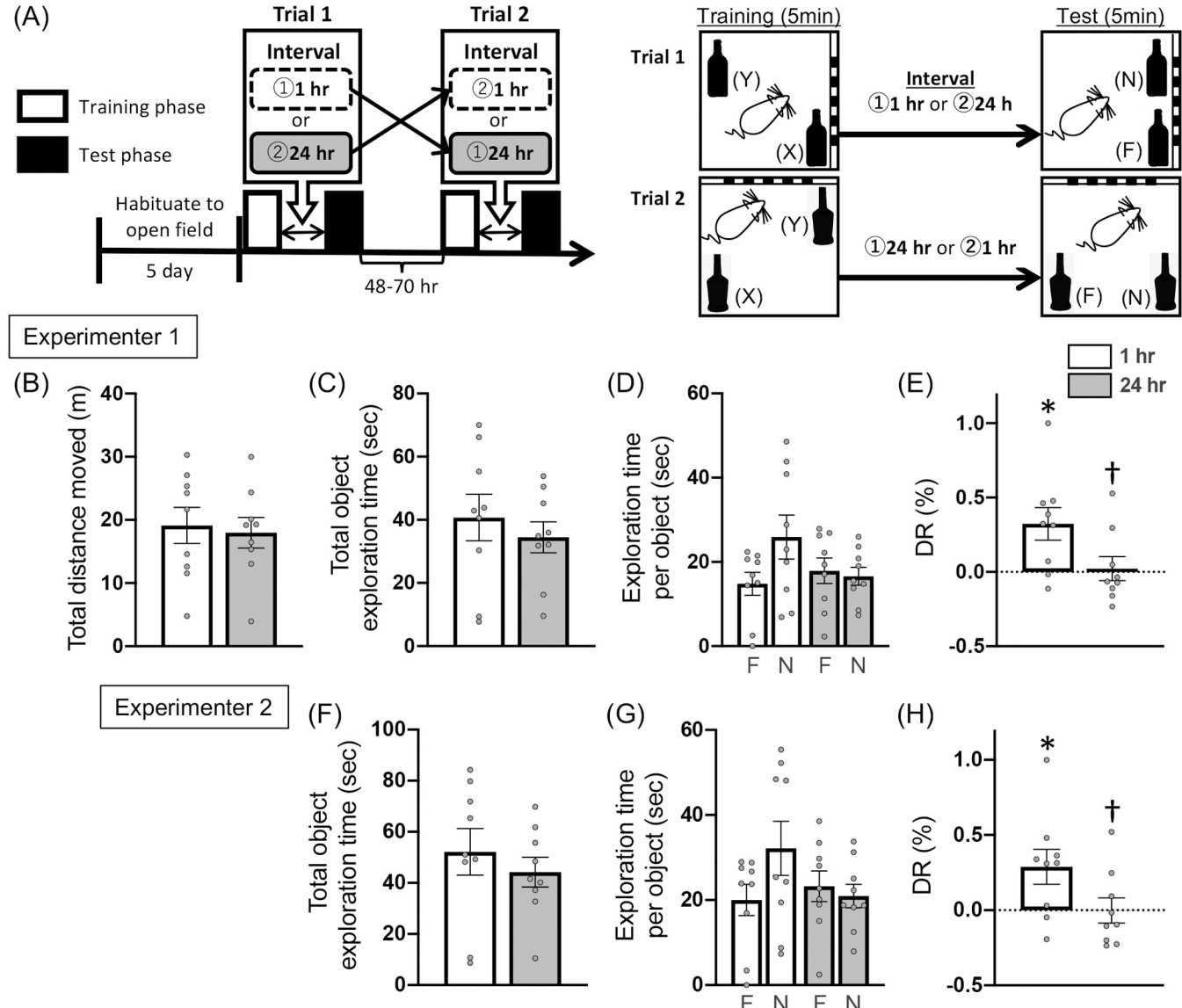

**Fig 3. Effects of different intervals on object location memory: 1-hr vs 24-hr retention intervals.** Expt. 1a: (A) Schematic timeline of the experimental procedures; (B) total distance moved; (C, F) total object exploration time; (D, G) exploration time of familiar [F] or novel [N] location objects; and (E, H) discrimination ratio [DR] for the test phase. The exploration behavior was analyzed by two independent experimenters. Panels (C)–(E) present the results obtained by Experimenter 1, whereas panels (F)–(H) show those obtained by Experimenter 2. White and gray columns indicate 1-hr and 24-hr retention interval conditions, respectively. Data are shown as the mean ± SE (n = 9). *$p < .05$ vs chance level (one-sample t-test). †$p < .05$ vs the 1-hr retention interval condition (paired t-test).

= 0.806, $p = .382$; $F_{ex×ob}(1,17) = 0.910$, $p = .353$; two-way RM ANOVA; Fig 4D]. DRs were significantly above chance level under the AME condition, but not under the Sed condition [AME: one-sample $t(17) = 2.384$, $p = .029$, Cohen's $d = .562$; Sed: one-sample $t(17) = 1.496$, $p = .153$; Fig 4E], despite there being no difference in the DRs between the two conditions [paired $t(17) = 1.024$, $p = .320$; Fig 4E].

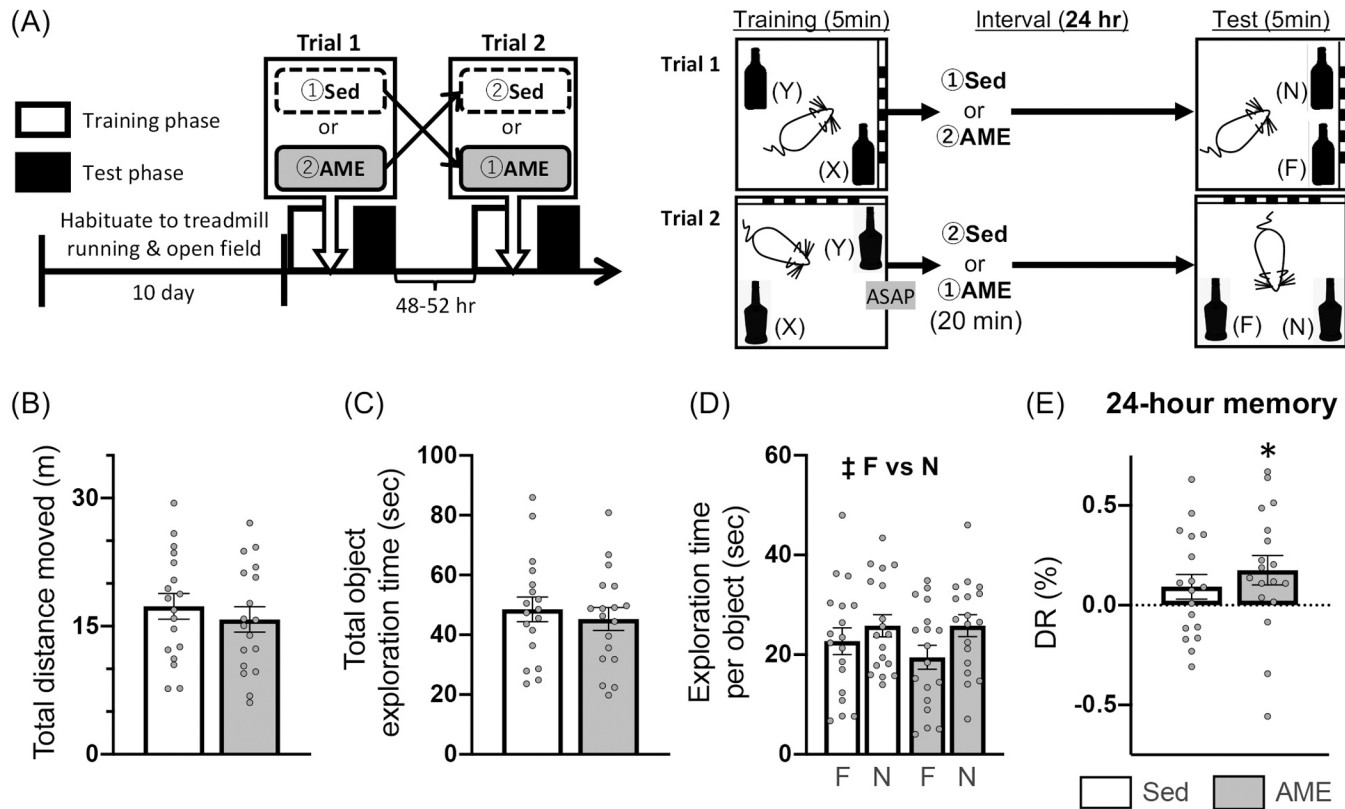

**Fig 4. Effects of acute moderate-intensity exercise immediately after object location learning.** Expt. 2: (A) Schematic timeline of the experimental procedures; (B) total distance moved; (C) total object exploration time; (D) exploration time of familiar [F] or novel [N] location objects; and (E) discrimination ratio [DR] for the test phase. White and gray columns indicate sedentary (Sed) and acute moderate-intensity exercise (AME) conditions, respectively. Data are shown as the mean ± SE (n = 18). ‡$p < .05$ main effect of object (F vs N) (two-way RM ANOVA). *$p < .05$ vs chance level (one-sample t-test).

## Combined effects of post-learning AME and a protein synthesis inhibitor on the OL test

We subsequently investigated whether the AME-induced enhancement of OL memory persistence is eliminated by an ANI injection into the dorsal hippocampus immediately after (Expt. 3) or 4 hr after (Expt. 4) the learning phase. The results of exploration behavior during the learning phase are summarized in C and D Table 3.

In the learning phase of Expt. 3 (C in Table 3), there were no significant interactions or main effects of AME intervention and ANI treatment on total object exploration time [$F_{ex×dr}(1,35) = 1.223$, $p = .276$; $F_{ex}(1,35) = 2.403$, $p = .130$; $F_{dr}(1,35) = 0.632$, $p = .432$; two-way mixed ANOVA], respective object exploration time [$F_{ex×dr×ob}(1,35) = 0.210$, $p = .649$; $F_{ob×dr}(1,35) = 0.079$, $p = .780$; $F_{ex×ob}(1,35) = 1.990$, $p = .327$; $F_{ob}(1,35) = 0.000$, $p = .995$; three-way mixed ANOVA], and DRs [$F_{ex×dr}(1,35) = 0.108$, $p = .774$; $F_{ex}(1,35) = 0.810$, $p = .374$, $F_{dr}(1,35) = 0.051$, $p = .823$; two-way ART ANOVA]. DRs also did not differ from chance levels in either group [Sed/Sal: one-sample $t(18) = 0.117$, $p = .909$; Sed/ANI: one-sample $t(18) = 0.147$, $p = 0.885$; AME/Sal: one-sample $t(17) = 0.539$, $p = 0.597$; Sed/ANI: one-sample $t(17) = 0.709$, $p = .488$]. By contrast, total distance moved showed a significant interaction between AME intervention and ANI treatment, with no main effects of the two factors [$F_{ex×dr}(1,35) = 5.269$, $p = .028$, $\eta^2_p = .131$; two-way mixed ANOVA]. In the AME group, the total distance moved under the ANI injection condition was significantly longer than under the Sal injection condition ($p = .013$; Bonferroni *post-hoc* test).

With the test phase, there was a significant main effect of ANI treatment, but not of AME intervention, and no significant the interaction on total distance moved [$F_{dr}(1,35) = 12.041$, $p = .001$, $\eta^2_p = .256$; $F_{ex}(1,35) = 0.026$, $p = .875$; $F_{ex \times dr}(1,35) = 1.484$, $p = .231$; two-way mixed ANOVA Fig 5B]. Specifically, the total distance moved in the ANI treatment condition was longer than that in the Sal treatment condition. Total object exploration time was not affected by the AME intervention or ANI treatment [$F_{ex \times dr}(1,35) = 0.296$, $p = .590$; $F_{ex}(1,35) = 0.000$, $p = .985$; $F_{dr}(1,35) = 2.236$, $p = .144$; two-way ART ANOVA; Fig 5C)], whereas respective object exploration time was significantly affected by ANI treatment, but not by the AME intervention or object [$F_{dr}(1,105) = 4.143$, $p = .044$; $F_{ex \times dr \times ob}(1,105) = 2.232$, $p = .138$; $F_{ex \times dr}(1,105) = 0.390$, $p = .533$; $F_{ex \times ob}(1,105) = 2.763$, $p = .099$; $F_{dr \times ob}(1,105) = 3.562$, $p = .062$; $F_{ex}(1,35) = 0.033$, $p = .857$; $F_{ob}(1,105) = 3.712$, $p = .057$; three-way ART ANOVA; Fig 5D]. ANI significantly increased respective object exploration time compared to Sal. Here, DRs exceeded the chance level only in the AME/Sal condition [AME/Sal: one-sample $t(17) = 3.980$, $p = .001$, Cohen's $d = .938$; Sed/Sal: one-sample $t(18) = 0.288$, $p = .776$; Sed/ANI: one-sample $t(18) = 0.849$, $p = .407$; AME/ANI: one-sample $t(17) = 0.320$, $p = .753$; Fig 5E]. Moreover, the DRs showed a significant interaction between the AME intervention and drug injection [$F_{ex \times dr}(1,35) = 10.724$, $p = .002$, $\eta^2_p = .235$; two-way mixed ANOVA; Fig 5E], with significantly higher DRs in the AME/Sal than in both the AME/ANI ($p = .000$) and Sed/Sal ($p = .004$) conditions (Bonferroni post-hoc test).

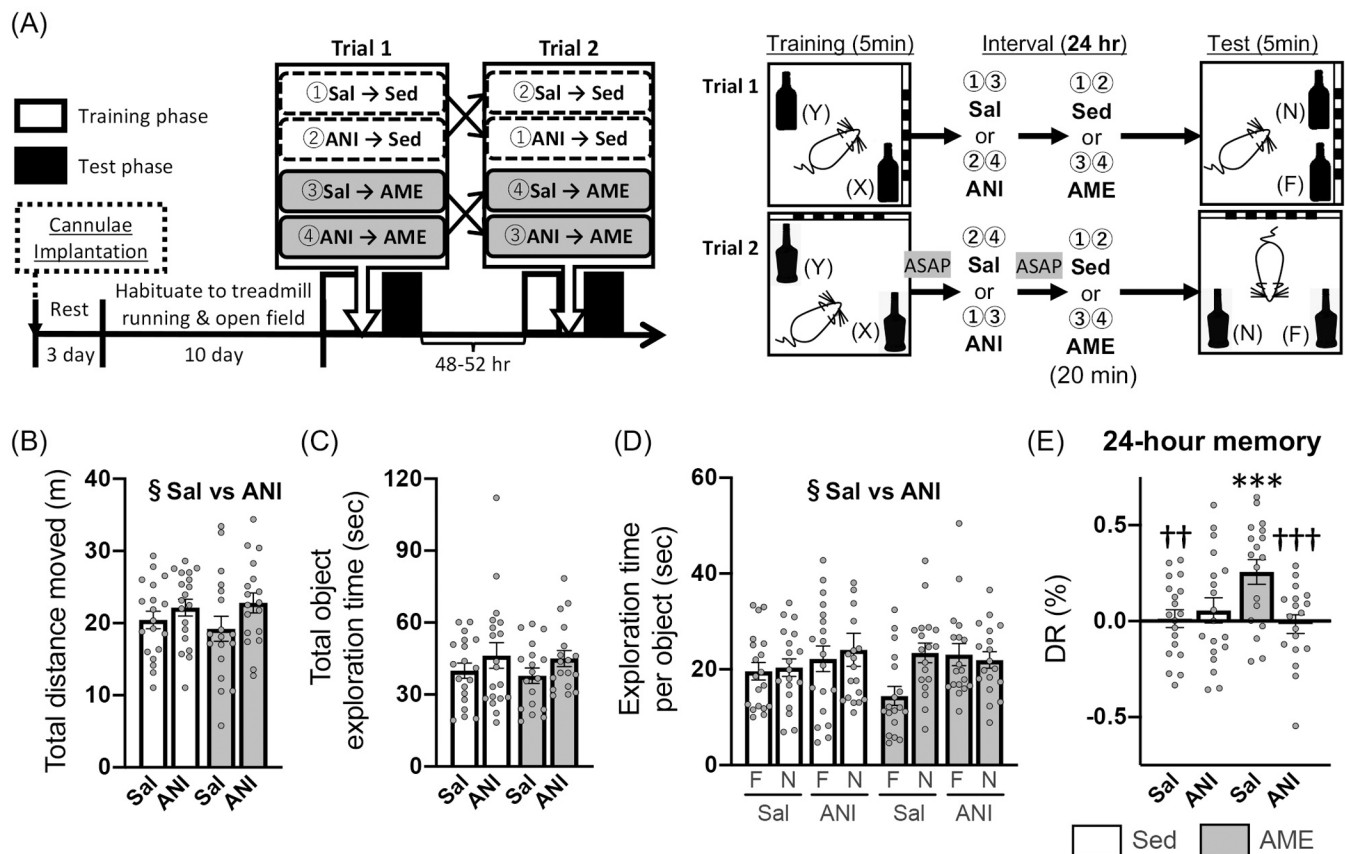

**Fig 5. Effect of protein synthesis inhibition immediately after learning on AME-induced enhancement of OL memory.** Expt. 3: (A) Schematic timeline of the experimental procedures; (B) total distance moved; (C) total object exploration time; (D) exploration time of familiar [F] or novel [N] location objects; and (E) discrimination ratio [DR] for the test phase. White and gray columns indicate Sed and AME conditions, respectively. Data are shown as the mean ± SE (Sed: n = 19, AME: n = 18). §$p < .05$ main effect of drug (Sal vs. ANI, two-way mixed ANOVA or three-way ART ANOVA). ***$p < .001$ vs chance level (one-sample t-test). ††$p < .01$, †††$p < .001$ vs AME/Sal (two-way mixed ANOVA with Bonferroni post-hoc test).

In Expt. 4, during the learning phase (D in Table 3), total distance moved was not affected by AME intervention and ANI treatment [$F_{ex \times dr}(1,27) = 0.198$, $p = .660$; $F_{ex}(1,27) = 0.303$, $p = .587$; $F_{dr}(1,27) = 0.576$, $p = .455$; two-way ART ANOVA]. In contrast, total and respective object exploration time showed a significant interaction between the intervention and the treatment [total: $F_{ex \times dr}(1,27) = 5.460$, $p = .027$, $\eta^2_p = 0.168$; $F_{ex}(1,27) = 1.403$, $p = .247$; $F_{dr}(1,27) = 291$, $p = .594$; two-way mixed ANOVA; respective: $F_{ex \times dr \times ob}(1,27) = 0.550$, $p = .465$; $F_{ex \times ob}(1,27) = 1.897$, $p = .180$; $F_{dr \times ob}(1,27) = 0.533$, $p = .472$; $F_{ob}(1,27) = 0.251$, $p = .621$, three-way mixed ANOVA], with significantly longer exploration time in the Sed/Sal condition than in the Sed/ANI ($p = .048$) and AME/Sal ($p = .018$) conditions (Bonferroni post-hoc test). However, there was no significant group differences in DRs between any of the conditions [$F_{ex \times dr}(1,27) = 0.035$, $p = .853$; $F_{ex}(1,27) = 1.998$, $p = .169$; $F_{dr}(1,27) = 0.593$, $p = .448$; two-way mixed ANOVA], and the DRs did not differ from the chance level under any condition [Sed/Sal: one-sample $t(14) = 0.227$, $p = .786$; Sed/ANI: one-sample $t(14) = 0.562$, $p = .583$; AME/Sal: one-sample $t(13) = 1.899$, $p = .080$; AME/ANI: one-sample $t(13) = 0.505$, $p = .622$].

For the test phase, total distance moved and total object exploration time were not significantly affected by either AME intervention, ANI treatment, or their interaction [distance: $F_{ex \times dr}(1,27) = 1.882$, $p = .181$; $F_{ex}(1,27) = 0.991$, $p = .328$; $F_{dr}(1,27) = 1.447$, $p = .239$; two-way mixed ANOVA; Fig 6B; exploration: $F_{ex \times dr}(1,27) = 0.016$, $p = .900$; $F_{ex}(1,27) = 0.869$, $p = .359$;

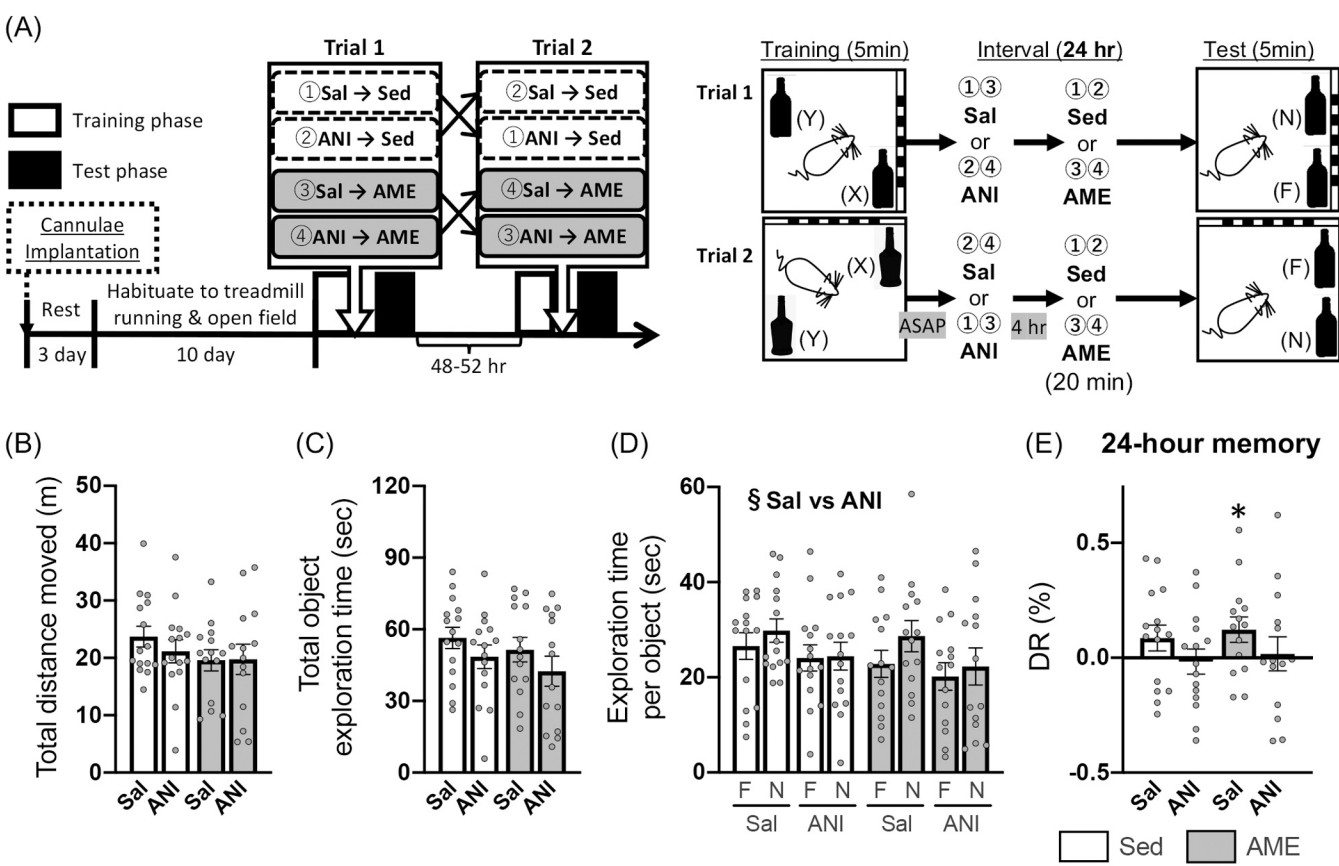

**Fig 6. Effect of protein synthesis inhibition 4 hr after learning on AME-induced enhancement of OL memory.** Expt. 4: (A) Schematic timeline of the experimental procedures; (B) total distance moved; (C) total object exploration time; (D) exploration time of familiar [F] or novel [N] location objects; and (E) discrimination ratio [DR] for the test phase. White and gray columns indicate exercise (AME) and sedentary (Sed) conditions, respectively. Data are shown as the mean ± SE (Sed: n = 15, AME: n = 14). §$p < .05$ main effect of drug (Sal vs ANI; three-way ART ANOVA). *$p < .05$ vs chance level (one-sample t-test).

$F_{dr}(1,27) = 3.590$, $p = .069$; two-way mixed ANOVA; Fig 6C]. In contrast, respective object exploration time showed a significant main effect of drug, with the exploration time in the Sal treatment condition being significantly longer than that in the ANI treatment condition [$F_{ex \times dr \times ob}(1,81) = 0.056$, $p = .813$; $F_{ex \times dr}(1,27) = 0.009$, $p = .927$; $F_{ex \times ob}(1,27) = 0.121$, $p = .729$; $F_{dr \times ob}(1,27) = 0.580$, $p = .448$; $F_{ex}(1,27) = 0.986$, $p = .330$; $F_{dr}(1,27) = 5.601$, $p = .020$; $F_{ob}(1,27) = 1.841$, $p = .179$; three-way ART ANOVA; Fig 6D]. Furthermore, despite the absence of significant differences in DRs across conditions [$F_{ex \times dr}(1,27) = 0.000$, $p = .984$; $F_{ex}(1,27) = 0.471$, $p = .499$; $F_{dr}(1,27) = 2.349$, $p = .137$; two-way mixed ANOVA, Fig 6E], the AME/Sal condition showed a DR significantly above the chance level [AME/Sal: one-sample $t(13) = 2.214$, $p = .045$, Cohen's $d = .592$; Sed/Sal: one-sample $t(14) = 1.535$, $p = .147$; Sed/ANI: one-sample $t(14) = 0.306$, $p = .764$; AME/ANI: one-sample $t(13) = 0.234$, $p = .819$; Fig 6E].

### Effects of repeated OL test exposure

In all experiments, each rat underwent two OL test trials (learning phase + test phase) under different experimental conditions. To assess whether repeated exposure to the OL test influenced exploratory behavior, we compared total distance moved and total object exploration time between the 1st and 2nd OL test trials. Results are summarized in Table 4.

Total distance moved during the learning and test phases did not significantly differ between the first and second trials in any of the experiments (A–D in Table 4). On the other hand, total exploration time in each experiment showed a significant difference between the two trials during the OL learning phase, the test phase, or both. In Expt. 1a, total exploration time during the learning phase did not differ between the 1st and 2nd OL trials, while the time during the test phase was longer in the 2nd trial than in the 1st one [learning: experimenter 1; paired $t(8) = .961$, $p = .365$; experimenter 2; paired $t(8) = 1.194$, $p = .267$; test: experimenter 1; paired $t(8) = 2.809$, $p = .023$, Cohen's $d = -.936$; experimenter 2; $w(9) = 39.00$,

**Table 4. Behavioral measures of the first and second OL tests.**

| Experiment | Trial | Total distance moved (m) | | Total exploration time (s) | |
|---|---|---|---|---|---|
| | | Learning | Test | Learning | Test |
| A. Expt. 1a (n = 9) | | | | | |
| Experimenter 1 | First | 20.53 ± 1.26 | 16.38 ± 2.33 | 36.32 ± 4.21 | 27.58 ± 4.93 |
| | Second | 18.10 ± 2.04 | 20.72 ± 2.73 | 47.26 ± 8.03 | 47.61 ± 5.59* |
| Experimenter 2 | First | | | 44.04 ± 5.04 | 35.01 ± 6.45 |
| | Second | | | 58.51 ± 8.40 | 61.35 ± 6.02# |
| B. Expt. 2 (n = 18) | | | | | |
| | First | 16.86 ± 1.11 | 16.31 ± 1.40 | 57.58 ± 4.70 | 48.42 ± 4.43 |
| | Second | 15.91 ± 1.78 | 16.80 ± 1.59 | 45.49 ± 5.06* | 45.38 ± 3.47 |
| C. Expt. 3 (Sed = 19, AME = 18) | | | | | |
| Sed | First | 22.60 ± 1.22 | 21.20 ± 1.24 | 59.57 ± 3.88 | 51.22 ± 4.94 |
| | Second | 22.64 ± 1.06 | 21.41 ± 1.17 | **48.10 ± 3.87** | **34.98 ± 2.82** |
| AME | First | 22.62 ± 1.13 | 21.97 ± 1.49 | 51.76 ± 3.83 | 45.94 ± 3.15 |
| | Second | 20.47 ± 1.66 | 20.06 ± 1.74 | **42.08 ± 3.04** | **36.94 ± 3.26** |
| D. Expt. 4 (Sed = 15, AME = 14) | | | | | |
| Sed | First | 22.45 ± 1.16 | 21.99 ± 2.17 | 60.35 ± 4.75 | 50.36 ± 5.39 |
| | Second | 23.44 ± 1.86 | 22.89 ± 1.68 | **53.86 ± 3.32** | 54.55 ± 4.10 |
| AME | First | 22.13 ± 1.94 | 18.73 ± 2.01 | 56.97 ± 5.24 | 43.68 ± 5.38 |
| | Second | 19.70 ± 2.06 | 20.61 ± 2.48 | **44.46 ± 3.76** | 50.24 ± 6.25 |

Sed: sedentary control; AME: acute moderate exercise. Data are expressed as mean ± SE, *$p < .05$ vs first trial by paired t-test. #$p < .05$ vs first trial by Wilcoxon signed-rank test. Bold text shows a significant main effect of trial ($p < .05$ vs first trial) by two-way mixed ANOVA.

$p = .020$; A in Table 4]. In Expt. 2, total exploration time during the learning phase was shorter in the 2nd trial than in the 1st, with no significant difference observed between them during the test phase [learning: paired $t(17) = 2.791$, $p = .012$, Cohen's $d = .658$; test: paired $t(17) = .846$, $p = .409$; B in Table 4]. In Expt.3, total exploration time in the 2nd trial was significantly shorter than in the 1st during both learning and test phases, regardless of the AME intervention [learning: $F_{ex×tr}(1,35) = 0.111$, $p = .741$; $F_{ex}(1,35) = 2.403$, $p = .130$; $F_{tr}(1,35) = 15.375$, $p = .000$, $\eta^2_p = .305$; two-way mixed ANOVA; test: $F_{ex×tr}(1,35) = 1.537$, $p = .233$; $F_{ex}(1,35) = 0.151$, $p = .700$; $F_{tr}(1,35) = 18.660$, $p = .000$, $\eta^2_p = .348$; two-way mixed ANOVA; C in Table 4]. In Expt.4, a significantly shorter total exploration time was found in the 2nd trial compared to the 1st during the learning phase, but not during the test phase [learning: $F_{ex×tr}(1,35) = 1.403$, $p = .247$; $F_{ex}(1,35) = 0.486$, $p = .490$; $F_{tr}(1,35) = 10.898$, $p = .003$, $\eta^2_p = .288$; two-way mixed ANOVA; test: $F_{ex×tr}(1,27) = 0.065$, $p = .800$; $F_{ex}(1,27) = 0.869$, $p = .359$; $F_{tr}(1,27) = 1.343$, $p = .257$; two-way mixed ANOVA; D in Table 4].

## Discussion

This study tested the hypothesis that AME after learning enhances memory consolidation in the hippocampus-dependent OL test, promoting greater OL memory persistence via hippocampal protein synthesis, especially in its dorsal area. To do this, we first examined whether OL memory, which is typically lost within 24h, could be retained beyond 24 hr by implementing AME immediately after learning (Expt. 1a & 2). We then investigated whether the post-learning AME-induced improvement in retention of OL memory is eliminated with an injection of the protein synthesis inhibitor anisomycin (ANI) into the dorsal hippocampus (Expt. 3 & 4). We found that implementing AME following learning enhanced OL memory persistence and that this enhancement was suppressed when the protein synthesis in the dorsal hippocampus was inhibited. These findings supported our hypothesis, indicating that the AME-induced consolidation of OL memory depends on *de novo* protein synthesis in the dorsal hippocampus, and underscore the effectiveness of moderate-intensity exercise intervention during the consolidation phase for long-term memory (LTM) formation.

With Expt. 1a, we investigated whether an OL memory formed during a 5-min learning phase persists in the short term (1 hr after learning) but not in the long term (24-hr after learning). OL exploration behavior during the learning phase was no different between the two-retention interval conditions (A in Table 3), indicating that each object was learned to the same extent in both conditions. During the test phase, there was no difference in respective object exploration for either condition (Fig 3D), while DRs in the 1-hr condition were significantly above chance level and were higher than those in the 24-hr condition (Fig 3E). The reliability of the observed behavior results determined by an experimenter was supported by a second experimenter who independently analyzed the same behavioral data (video footage), showing a high inter-rater agreement between the two experimenters across both phases (Section of results for Expt. 1a). In spontaneous object memory tests, absolute exploration time is influenced by individual variability in exploration levels, and thus relative indices such as the DR, which can control for this variability, are considered more reliable measures of discrimination ability, and by extension, memory performance [51]. Given that, our findings indicate that the OL memory formed during the 5 min of learning could be retained for 1 h, but not for 24 h. Similar results have been reported in past studies on rats [23] and mice [22], indicating that an OL memory acquired through 5 min of learning persists for 1–6 h, but not for 24 h.

Under the 24-hr retention interval condition, in Expt. 2, AME intervention immediately after learning led to significantly higher DRs than chance level for the test phase, with no difference in the DRs of the Sed condition (Fig 4E). Similarly, in Expt. 3, DRs elevated relative to chance by AME intervention were observed when saline (Sal) was administered into the dorsal hippocampus immediately after learning, whereas such enhancement was not observed when ANI was administrated (Fig 5E). At this time, the DRs in the AME/Sal group were significantly higher than those in the other groups. On the other hand, in Expt. 4 where the dorsal hippocampal injection of Sal or ANI were performed four hours after learning, significantly higher DRs than chance level were found only in the AME/Sal group, and no differences were observed between these DRs and those of other groups (Fig 6E). It is noteworthy that the DRs consistently exceeded chance level with the post-learning AME intervention, with medium-to-large effect size in Experiments 2−4 (Expt. 2: Cohen's $d = .562$; Expt. 3: Cohen's

*d* = .938; Expt. 4: Cohen's *d* = .592). In the spontaneous object memory task, comparing DRs between groups is based on the assumption that their magnitude reflects memory strength. This assumption has been questioned in previous studies, which suggest that assessing whether DRs significantly differ from chance level may provide a more accurate indication of the presence or absence of memory [52]. The reason for which no significant differences in DRs were observed between the AME or AME/Sal groups and the other groups remains unclear. Nonetheless, taken together, our results indicate the following: (1) AME intervention immediately after 5 min of learning enhances OL memory consolidation, resulting in memory retention 24 hours later and (2) this enhancement is abolished by the dorsal hippocampal injection of ANI immediately after learning and may also be suppressed when ANI is injected 4 hours after learning.

This study was designed for each rat to perform two OL tests under different experimental conditions. Although repeated exposure to the OL test did not affect general activity levels, object exploration behavior differed between the 1st and 2nd OL test trials (Table 4). Specifically, total exploration time tended to increase with repeated testing in Expt. 1a, whereas it tended to decrease in Expt. 2–4. The reason for the increased exploratory behavior in the 2nd trial of Expt. 1a remains unclear. In contrast, the reduction in exploratory behavior with repeated testing observed has also been reported in other behavioral paradigms, such as the object recognition (OR) test [53] and the open field (OF) test [54]. As previously noted, the DR is a relative index that accounts for differences in exploration levels [51]. Differences in exploration behavior were observed between the two repeated OL test trials; nevertheless, interpreting the results based on DR remains valid and appropriate.

To our knowledge, this is the first study to demonstrate that post-learning moderate-intensity exercise promotes the consolidation of OL memory in rodents. Past studies using the OR test have reported that exercise immediately after learning has a comparable promoting effect on the persistence of an OR memory in rats [27–31]. It is generally accepted that OR memory primally relies on the perirhinal cortex (PRh) [32,33], whereas the role of the hippocampus in OR memory remains under debate [55,56]. On the other hand, OL memory depends on the hippocampus, particularly the dorsal area [57]. Given this, the AME-enhanced memory consolidation model proposed in the present study may serve as a more effective and stronger tool for elucidating the hippocampus-centered mechanisms underlying the enhancement of memory consolidation with exercise, compared to models based on the OR test used in previous studies. In addition, previous studies with humans have reported enhanced consolidation of associative [26], spatial [58], vocabulary [59] and picture-location [25] memories with post-learning exercise. Among these human studies, most have reported that exercise immediately or within an hour after learning improves memory retrieval after 24 hr or more [26,58,59], similar to the present study. Conversely, one study indicated no benefit when exercise was performed immediately after learning but did observe memory improvement when exercise was performed 4 hr after learning [25]. This partial discrepancy is thought to be due to differences in experimental conditions such as the subjects, the exercise, and the memory task. Importantly, all of the studies, including ours, show the effectiveness of exercise intervention during the consolidation phase of memory retention, supporting the previous meta-analysis [6].

The impact of acute exercise during the consolidation phase on memory retention varies with exercise intensity, and vigorous exercise is considered the most effective [6]. For instance, exercise at an intensity of 60–70% maximal oxygen uptake ($VO_{2max}$) after learning enhances the persistence of an OR memory in rats [27–31]. This intensity corresponds to the vigorous-exercise zone as classified in the American College of Sports Medicine (ACSM) guidelines [60,61]. By contrast, the running speed of LT applied in this study is located at the 41.0–63.6% of $VO_{2max}$ [43], which is equivalent to the moderate-intensity zone as classified in the ACSM guidelines [60,61]. This study therefore shows that even moderate-intensity exercise, that is exercise, which is slightly lighter than vigorous exercise, produces a sufficient effect for promoting memory consolidation. This finding is supported by another human study, which shows that the associative memory task performance can be improved by having subjects perform moderate exercise after learning, but not by having them perform more intense exercise [26]. The combined results of the present study and this previous study suggest that moderate-intensity exercise, which is safer and more accessible than vigorous exercise, has the potential to enhance memory consolidation.

Previously, a novel OF environment or electric foot shocks were reported as a stimulus that enhances memory consolidation and triggers LTM formation in hippocampus-dependent memory tasks. For example, an inhibitory avoidance (IA) task combined with novel OF exposure, but not with familiar OF exposure, leads to the consolidation of the transient IA memory into a long-term IA memory [7]. Similar effects that enhance memory consolidation through post-learning novel OF exposure have been confirmed for various tasks, such as the contextual fear conditioning task, the conditioned taste aversion task, the OR task, the unique everyday memory task and the water maze task [11,62–64]. On the other hand, foot shocks have long been used as an emotionally arousing unconditioned stimulus that enhances memory consolidation, in various memory tasks [65]. Furthermore, conditioning stimuli or environments paired with foot shocks have been reported as enhancing agents for memory consolidation when implemented after learning [66,67]. It should be noted that, in this study, the effects of AME on OL memory were tested after habituation to the OF environment and treadmill running. During the OL test, rats in the AME group ran voluntarily without foot shocks, while rats in the Sed group rested on the treadmill for the same duration of time as the AME rats. Therefore, the post-learning AME-induced enhancement of memory consolidation in this study was likely unaffected by environmental novelty from the OF or by foot shocks.

The enhanced OL memory persistence with post-learning AME was eliminated with the administration of the protein synthesis inhibitor ANI into the dorsal hippocampus immediately after learning (Fig 5E) and was possibly suppressed when ANI was administered 4 hours later (Fig 6E). These findings indicate that *de novo* protein synthesis in the dorsal hippocampus is essential for enhancing memory persistence by post-learning AME intervention. More specifically, sustained hippocampal protein synthesis initiated immediately after AME, as well as its delayed upregulation occurring several hours later, may contribute to the enhancement of memory retention by AME. The protein synthesis-dependent enhancement of memory consolidation by AME are consistent with the framework of the BT model [7], which is a behavioral analog of the synaptic tagging and capture (STC) model [68,69]. With the BT model, a weak learning task that usually forms STM converts to LTM through a mechanism dependent upon protein synthesis when an unrelated strong or novel behavioral event occurs shortly before or after the learning phase. Indeed, previous studies using hippocampus-dependent tasks have shown that spatial novelty can promote the consolidation of STM generated by weak learning into LTM when the novelty is introduced at a timing similar to the "memory's penumbra" observed in this study, while the effects were abolished by the injection of ANI into the dorsal hippocampus [7,11,12,63]. From these results, it can be seen that the BT model proposes that weak learning sets a "behavioral tag" at specific synaptic sites, while a concurrent behavioral event supplies *de novo* plasticity-related proteins (PRPs) that the tags capture, resulting in the conversion of a weak memory into a LTM [70]. Importantly, the behavioral tag is set through a protein-synthesis-independent process [11]. Previous studies using the OL test have shown that weak encoding, consisting of a single learning phase for approximately 5–10 minutes leads to the formation of STM lasting up to about 6 hours but does not result in LTM retention after 24 hours [23,71,72]. These findings indirectly suggest that a single 5-min OL learning, as used in this study, serves as a stimulus that does not involve *de novo* protein synthesis. In contrast, AME implemented immediately after OL learning activates the dorsal hippocampus, accompanied with the induction of several PRPs, such as BDNF, Arc, and mTOR [34,73–75]. Based on the BT model, the weak encoding by 5-min OL learning may set synaptic tags, and the subsequent AME supplies the PRPs that are essential for capturing these tags and stabilizing the memory, thereby enabling LTM formation. However, since this study did not examine changes in total or specific protein levels, it remains unclear which specific PRP or PRPs act as key regulators in the successful formation of BT through exercise, a topic for future research.

Additionally, in the BT model, converting a weak learning memory into LTM through a behavioral event is believed to require an overlap in the brain regions—or more precisely, the neuronal populations—activated by the learning and behavioral events. For example, a novel spatial environment that activates the hippocampus, but not a novel taste that activates the insular cortex, promotes the consolidation of transient hippocampus-dependent STM into LTM in the OR and IA tasks [7,11]. Another study has shown that STMs generated by weak OR learning are converted into OR-LTM through pre- or post-exposure to a novel spatial context (an unfamiliar chamber), during which shared neuronal ensembles

corresponding to both OR learning and spatial novelty were activated in the dorsal hippocampus [12]. Also, optical silencing of neuronal ensembles corresponding to the spatial novelty suppressed the formation of OR-LTM. The BT is thus achieved through the activation of a common neuronal ensemble in the hippocampus. In the present study, we found that post-learning AME could enhance the consolidation of OL-STMs into LTMs via protein synthesis in the dorsal hippocampus. This hippocampal region plays a crucial role in the OL task [19–21] as well as being activated by moderate-intensity exercise [34]. These findings suggest that shared neuronal populations in the dorsal hippocampus might contribute to memory consolidation via post-learning moderate-intensity exercise. Furthermore, an interesting finding has been reported showing that post-learning exercise improves memory retention in a hippocampus-dependent task and increases hippocampal pattern similarity for correct answers during memory recall [25]. This finding implies that exercise may enhance synaptic connectivity among engram cell ensembles associated with the memory task. In any case, whether the exercise-induced enhancement of memory consolidation stems from the selective activation of neuronal populations shared between exercise and the memory task, or from overlapping activation as part of a broader hippocampal response, remains an intriguing question for future investigation.

Exercise at the running speed of low- to high-intensity stimulates neuronal activity of the dorsal hippocampus with increased levels of monoamine, such as noradrenaline (NA) and dopamine (DA), which are associated with the activation of the locus coeruleus (LC) and ventral tegmental area (VTA) [38]. Past studies using the OR task have reported mono-aminergic activation in this region [29–31]; also, neuronal input from the LC region [28] plays an important role in the post-learning exercise-induced memory persistence. For example, high-intensity exercise after OR training promotes the persistence of OR memory with increased NA and DA levels in the hippocampus, and the administration of these receptors' antagonist eliminate the memory persistent effects [29–31]. Also, injection of the neuronal inhibitor muscimol into the LC, but not the VTA, suppresses OR memory persistence triggered by the implementation of high-intensity exercise after learning [28]. Importantly, existing evidence for the BT model has shown that the transition of hippocampus-dependent STM into LTM triggered by novelty is prevented when a DA receptor antagonist or protein synthesis inhibitor is injected into the dorsal hippocampus before exposure to the novelty [7,63,76,77]. Other studies have reported that DA enhances protein synthesis in the hippocampal neurons, or in the dorsal area of the hippocampus, which enables DA-dependent neuronal activity, especially long-term potentiation (LTP) [78,79]. In short, novelty-induced boost of memory consolidation may be mediated by DA-dependent protein synthesis in the hippocampus. Given these facts, the dorsal hippocampal activation via monoaminergic input from the LC or VTA regions may contribute to the protein synthesis-dependent transition from OL-STM to LTM by post-leaning AME observed in this study.

## Conclusion

Taken together, our results provide new evidence that a single bout of moderate exercise after learning boosts consolidation of object location memory, resulting in support for prolonging memory retention through *de novo* protein synthesis in the dorsal hippocampus. This finding suggests that exercise can serve as a trigger to transform weakly encoded short-term memories into long-term memories within the framework of the "behavioral tagging" hypothesis [7]. It also highlights the crucial role of new protein synthesis in the hippocampus in regulating this exercise-induced enhancement of memory consolidation, laying the foundation for future studies to elucidate the underlying mechanisms in greater detail.

## Supporting information

**S1 Table. Behavioral data from Exp. 1a for each rat.**
(DOCX)

**S2 Table. Behavioral data from Exp. 2 for each rat.**
(DOCX)

**S3 Table. Behavioral data from Exp. 3 for each rat.**
(DOCX)

**S4 Table. Behavioral data from Exp. 4 for each rat.**
(DOCX)

## Acknowledgments

We are grateful to the laboratory members for analyzing the behavioral data and for their supportive discussion.

## Author contributions

**Conceptualization:** Koshiro Inoue, Masahiro Okamoto.

**Data curation:** Koshiro Inoue, Masahiro Okamoto, Takemune Fukuie.

**Formal analysis:** Koshiro Inoue.

**Funding acquisition:** Koshiro Inoue, Masahiro Okamoto, Hideaki Soya.

**Investigation:** Koshiro Inoue, Takemune Fukuie, Akihiko Yamaguchi.

**Methodology:** Koshiro Inoue, Masahiro Okamoto, Takemune Fukuie, Hideaki Soya.

**Resources:** Akihiko Yamaguchi.

**Supervision:** Hideaki Soya, Akihiko Yamaguchi.

**Validation:** Koshiro Inoue.

**Visualization:** Koshiro Inoue.

**Writing – original draft:** Koshiro Inoue.

**Writing – review & editing:** Masahiro Okamoto, Takemune Fukuie, Hideaki Soya, Akihiko Yamaguchi.

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
