## [Decision Letter · Decision Letter 0]

Dear Dr. Inoue,

We look forward to receiving your revised manuscript.

Kind regards,

Etsuro Ito, Ph.D.

Academic Editor

PLOS ONE

“This study was supported by grants from Japan Society for the Promotion of Science (26750307, 23K10637), and a grant from Advanced Research Initiative for

Human High Performance (ARIHHP), University of Tsukuba.”

Reviewers' comments:

Reviewer's Responses to Questions

**Comments to the Author**

1. Is the manuscript technically sound, and do the data support the conclusions?

Reviewer #1: Yes

Reviewer #2: Partly

2. Has the statistical analysis been performed appropriately and rigorously?

Reviewer #1: Yes

Reviewer #2: No

3. Have the authors made all data underlying the findings in their manuscript fully available?

Reviewer #1: Yes

Reviewer #2: Yes

4. Is the manuscript presented in an intelligible fashion and written in standard English?

Reviewer #1: Yes

Reviewer #2: Yes

Reviewer #1: In this study, the authors demonstrate that post-learning acute moderate-intensity exercise (AME) enhances memory retention, and that this enhancement is abolished by the inhibition of de novo protein synthesis. The experimental design and methodology are appropriate, and the findings are highly intriguing. However, several issues should be addressed before the manuscript can be accepted for publication:

1. Inter-rater reliability: In the evaluation of exploration time using video observation, the authors should report inter-rater reliability (e.g., agreement rate or intraclass correlation coefficient) across multiple observers.

2. Learning strength, memory persistence, and protein synthesis: The study reports that 5 minutes of learning is insufficient to support memory retention after 24 hours. The authors should clarify why such weak encoding fails to induce long-term memory. Is it due to the absence of de novo protein synthesis? Previous studies have shown that repeated or stronger learning sessions can result in long-lasting memory formation. Since the central claim of this study is that AME enhances memory by promoting protein synthesis in the hippocampus, the authors should also discuss how the strength or duration of learning affects the initiation of protein synthesis and subsequent memory consolidation.

3. Timing of anisomycin (ANI) administration in Experiment 3 (Figure 1): In Experiment 3, ANI is administered just before AME. Is this correct? Shouldn’t ANI be administered immediately after AME rather than before it? If the aim is to demonstrate that memory enhancement is mediated by AME-induced protein synthesis, the timing of ANI administration should be based on the AME session rather than the learning session. While the actual timing may be the same, this should be clearly described to avoid conceptual confusion.

4. Hippocampal involvement in object recognition tasks: As the authors note, previous studies have reported that AME also enhances memory in object recognition tasks. However, it is generally considered that the hippocampus is not required for object recognition memory. How do the authors interpret this? Given that object location tasks (which involve spatial components) are typically associated with the dorsal hippocampus, whereas object recognition tasks are not, the distinction between these two types of tasks should be more clearly addressed.

Reviewer #2: The content of this manuscript addresses an important and interesting topic, specifically:

(1) A single bout of moderate-intensity exercise immediately following weak encoding in an object location task enhances memory persistence for at least 24 hours.

(2) Injection of anisomycin into the dorsal hippocampus immediately or 4 hours after encoding blocks the beneficial effects of moderate-intensity exercise on memory persistence.

The experimental workload is substantial, and overall, the manuscript demonstrates high quality. However, several critical issues and missing details must be addressed prior to publication.

–Major Issues–

[1] CA1 Region

The authors injected 2 µL of anisomycin (lines 156-168). It is difficult to believe that such a volume would remain restricted to the CA1 hippocampal subregion, even if injection cannula tips were accurately placed. Without data confirming anisomycin spread, conclusions regarding the specificity of anisomycin effects within the CA1 region are questionable. Please provide data or justification regarding drug diffusion within the CA1 region.

[2] Exploration Time per Object

The absolute exploration time per object alone is insufficient to accurately reflect cognitive ability, as it can be confounded by individual differences in general activity, motivation, or anxiety. Relative measures such as discrimination ratios would better represent true cognitive performance. Please address this limitation clearly.

[3] Statistical Analyses

[3.1] Number of Animals and Repeated Testing:

• Experimental designs depicted in Figure 1 and its legend are unclear regarding repeated testing. Were all animals tested multiple times?

• For Experiments 1 and 2, the number of samples shown in Figures 5 and 6 (19 and 15 samples, respectively) does not align with the reported total of 40 and 30 animals (line 133). Clarify whether animals underwent two or four tests. Additionally, repeated measures ANOVA is only appropriate if all animals experienced every condition; otherwise, a mixed-model analysis would be required. Please clarify these discrepancies and justify your choice of statistical method.

[3.2] Information in Figures and Legends:

Clearly indicate the timing and number of repeated tests in Figure 1 and legends of all relevant figures. Include explicit sample sizes (N numbers) for each group in all figures or legends.

[3.3] Validation of Behavioral Paradigm:

Given the high variability observed in total distance traveled and object exploration, analyze whether repeated exposure to the task affected behavior (e.g., reduced exploration due to familiarity or increased spatial understanding). If these analyses were performed previously, reference them clearly. Clarify protocols regarding repetition: were objects, arenas, or spatial cues changed? Was the interval between sessions consistently two days?

[3.4] Figure 4D, Lines 531-534: Since no significant difference between exercise and control groups was found, despite exercise group performance being significantly above chance, it remains unclear whether moderate-intensity exercise truly enhances memory retention. Adjust interpretations accordingly.

[3.5] Figure 6D, Line 537: Given the lack of significant differences in DRs (two-way repeated measures ANOVA), claiming that moderate-intensity exercise enhanced memory retention or that anisomycin injections at 4 hours blocked this effect seems unjustified. Clarify or revise these claims.

[4] Data Normality and Variance: All analyses assume data normality and homogeneity of variance. Explicitly state whether these assumptions were tested and include results in the methods section.

[5] Pre-experiment Preparations: Did the authors conduct power analyses or sample size calculations, or were group sizes determined based on literature or previous experiments? Clarify your approach.

[6] Data Presentation: Splitting the scoring duration (2, 3, 5 minutes) does not clearly add significant value. Such methodological detail may be better suited to supplementary materials unless strongly justified. Consider using only one duration (e.g., 5 minutes) for clarity in the main manuscript. If maintaining multiple durations, provide thorough justification and discuss implications explicitly.

[7] Results Presentation: Include exact p-values rather than "p < 0.05". Additionally, report non-significant findings clearly, especially if maintaining multiple scoring durations.

[8] Clarifying Methodology: The discrimination ratio (DR) described appears identical to the discrimination index (DI) as calculated in your reference 39, not DR in reference 20. Please clarify or correct the terminology used.

[9] Statistical Power: For all one-sample t-tests, include effect size data to support statistical interpretations.

[10] Memory's Penumbra, Line 122: Memory's penumbra timing varies by task. Ballarini et al. (PNAS, 2009) demonstrated a -1 to +2 hr beneficial effect window for novelty on the object location task, not extending to 4 hr. Clarify this discrepancy and discuss implications.

–Minor Issues–

• Line 62: Ref 6 does not explicitly define long-term memory (LTM). Please clearly define LTM as used in this manuscript.

• Line 68: Clearly define short-term memory (STM) as well.

• Line 70: Clarify or consider referring to Nomoto et al. for better explanation of memory engram overlap (https://www.nature.com/articles/ncomms12319).

• Line 77: Confirm if "behavioral action tag" is established terminology.

• Lines 116, 187, etc.: Correctly distinguish "trial" (single event) from "session" (group of trials). One session should consist of several trials such as encoding and recall.

• Line 128: Clearly state total animal numbers.

• Line 186: Explain conditions for conducting tests during the dark period, including light conditions (type, duration, intensity).

• Figures 4, 5, 6: Explicitly note "24-hour memory" in legends or figures.

• Lines 379-387: Confirm if this section describes encoding trials.

• Line 450: Correct Figure reference (Fig. 5 → Fig. 6).

• Lines 488-492: Confirm references 22 and 40 indeed assessed 24-hour memory; clarify if needed.

• Lines 508-510: Clarify the appropriateness of comparisons to human studies (refs 22, 40), particularly regarding 24-hour memory.

• Lines 559-577: The authors discuss the synaptic tagging and capture (STC) model here. However, they already discussed the behavioral tagging model in the previous paragraph. Given that the behavioral tagging model as a behavioral analog of the STC model, I feel that this paragraph is somewhat redundant.

• Please cite all supplemental materials in the main text.

• Discussion Suggestion: Consider discussing the conceptual discrepancy between novelty-induced memory enhancement (via spatial/environmental novelty and overlapping memory engrams) and exercise-induced enhancement. How does running induce overlap in memory engrams, if at all, or does exercise broadly stimulate neural activity without specific engram overlap?

**Do you want your identity to be public for this peer review?** For information about this choice, including consent withdrawal, please see our Privacy Policy

Reviewer #1: **Yes: ** Kazuo Yamada

Reviewer #2: **Yes: ** Kristoffer Højgaard, Tomonori Takeuchi

---

## [Author Response · Author response to Decision Letter 1]

25 Jun 2025

Response to Reviewers

RE: PONE-D-25-16208, "Memory persistence enhancement by post-learning moderate exercise requires de novo protein synthesis in the dorsal hippocampus" by Inoue, K, Okamoto, M, Fukuie, T, Soya, H, and Yamaguchi, A

We sincerely thank the two reviewers for their invaluable contributions to improving our manuscript. We carefully reviewed each comment and made necessary revisions accordingly. Below, we provide a comprehensive response addressing all points raised by the reviewers. We deeply appreciate the reviewers' time, effort, and attention to detail in evaluating our work. Their insightful feedback has significantly enhanced our manuscript, and we believe the revised version meets the standards of your esteemed journal.

Response to Reviewers

Reviewer #1

Comment 1: Inter-rater reliability: In the evaluation of exploration time using video observation, the authors should report inter-rater reliability (e.g., agreement rate or intraclass correlation coefficient) across multiple observers.

Answer 1: In this study, the same experimenter assessed the rats’ exploration behaviors throughout all experiments using video recordings. To confirm the reliability of the behavioral data analyzed by the experimenter, for the revised manuscript, a different experimenter performed a second analysis of the behavioral data using the same video recordings (Experiment 1a), thereby allowing us to assess the consistency (inter-rater reliability) between the two behavioral-analysis datasets. The behavioral analyses by both experimenters yielded similar results, and the intraclass correlation coefficients (ICC) 2,1 had excellent agreement (0.861 to 0.959) for both datasets, indicating the high reliability of the behavioral data analyzed for this study. For details, please refer to the Methods (ll. 256-258) and Results (ll. 359-363) section for Experiment 1a.

Comment 2: Learning strength, memory persistence, and protein synthesis: The study reports that 5 minutes of learning is insufficient to support memory retention after 24 hours. The authors should clarify why such weak encoding fails to induce long-term memory. Is it due to the absence of de novo protein synthesis? Previous studies have shown that repeated or stronger learning sessions can result in long-lasting memory formation. Since the central claim of this study is that AME enhances memory by promoting protein synthesis in the hippocampus, the authors should also discuss how the strength or duration of learning affects the initiation of protein synthesis and subsequent memory consolidation.

Answer 2: To the best of our knowledge, there is not yet sufficient evidence to robustly elucidate the learning stimuli conditions (e.g., duration or frequency) required to promote hippocampal protein synthesis, which underlies memory consolidation in the OL test. Furthermore, since this study did not examine changes in total or specific protein levels, it is difficult to determine whether the failure to form long-term memory with 5 minutes of learning was due to an absence or an insufficient level of de novo protein synthesis. In contrast, the relationship between learning strength and memory persistence has been investigated in several previous studies using the OL test. For example, weak encoding, consisting of a single learning phase for approximately 5 to 10 minutes, leads to the formation of short-term memory lasting up to about 6 hours, but does not result in long-term memory retention after 24 hours (Wally et al., 2022; Bayraktar 2021; Ozawa et al., 2011). To support long-term memory formation, strong encoding is required, either through a sequential learning paradigm or an extended learning period (e.g., 15–20 minutes) (Bayraktar 2021; Shimoda et al., 2021). Considering that protein synthesis is essential for long-term memory formation (Ozawa et al., 2014), these findings indirectly suggest that a single 5- to 10-min learning intervention, as used in the present study, does not induce hippocampal protein synthesis to a level sufficient for enhancing memory consolidation.

Importantly, in the behavioral tagging (BT) model, synaptic tags have been set through protein synthesis independent process (Ballarini et al., 2009). Moreover, moderate-intenisty exercise causes the induction of several plasticity-related proteins (PRPs), such as BDNF, Arc, and mTOR, in the hippocampus (Soya et al., 2007; Rahmi et al., 2024; Lyoyd et al., 2017; Venezia et al., 2017). Applying these points to the BT model, weak encoding with 5 minutes of OL learning may establish synaptic tags, and subsequent AME may provide PRPs, which are essential for capturing the tags and stabilizing memory, resulting in long-term memory formation.

These matters have been added to the Discussion section (ll. 692-706).

Comment 3: Timing of anisomycin (ANI) administration in Experiment 3 (Figure 1): In Experiment 3, ANI is administered just before AME. Is this correct? Shouldn’t ANI be administered immediately after AME rather than before it? If the aim is to demonstrate that memory enhancement is mediated by AME-induced protein synthesis, the timing of ANI administration should be based on the AME session rather than the learning session. While the actual timing may be the same, this should be clearly described to avoid conceptual confusion.

Answer 3: In this study, ANI was injected immediately after the learning phase in Experiment 3, and 4 hours after the learning phase in Experiment 4. These timelines were based on the timing of memory encoding in the OL test. Accordingly, when based on the timing of the AME intervention, the ANI was injected either immediately before or 3 hours and 40 minutes after the AME session. There is no discrepancy in this regard.

We have previously reported that exercise immediately activates the hippocampus, including enhanced cerebral blood flow (Nishijima et al., 2006), increased c-Fos expression (Soya et al., 2007), and monoaminergic activation (Hiraga et al., 2025). These findings indicated that exercise can rapidly influence the hippocampus shortly after its initiation, which was one reason for administering ANI immediately before the AME intervention to fully suppress the AME-induced protein synthesis. Furthermore, exercise leads to the induction of biphasic protein synthesis; for example, BDNF, one of the factors considered to enhance memory consolidation, is reported to be markedly induced beginning 2 to 6 hours after exercise. ANI was thus injected 4 hours after learning in order to suppress this delayed phase of protein synthesis. The rationale for ANI injected at these time points has been included in the final paragraph of the Introduction (ll. 130-136).

The results of our study demonstrate that ANI injection at any of the tested time points abolished the memory-enhancing effects of AME, indicating that hippocampal protein synthesis is essential for the AME-induced enhancement of memory persistence. Therefore, the main conclusions of this study remain valid.

Comment 4: Hippocampal involvement in object recognition tasks: As the authors note, previous studies have reported that AME also enhances memory in object recognition tasks. However, it is generally considered that the hippocampus is not required for object recognition memory. How do the authors interpret this? Given that object location tasks (which involve spatial components) are typically associated with the dorsal hippocampus, whereas object recognition tasks are not, the distinction between these two types of tasks should be more clearly addressed.

Answer 4: In general, the hippocampus (Hip) plays a key role in regulating OL memory (Vogel-Ciernia et al., 2014), while the perirhinal cortex (PRh) is primarily involved in OR memory (Warburton, 2015; Winters et al., 2008). However, the involvement of the Hip in OR memory remains a matter of debate, as some studies have reported hippocampal contributions to OR memory as well (Cinalli et al., 2020; Cohen et al., 2013). Given this, the enhancement of OR memory by exercise, which has been reported in previous studies, may be attributed to improved function of the Hip, PRh, or both. However, the current study specifically focuses on hippocampus-dependent OL memory, and addressing the effects of exercise on OR memory or the role of the Hip in OR memory is beyond its scope. Therefore, in the revised manuscript, we have included a brief discussion highlighting the differences in brain regions involved in OL and OR memory, and we clarify the relevance and significance of our findings in this context (ll. 619-626).

Response to Reviewers

Reviewer #2

Comment 1: [1] CA1 Region: The authors injected 2 µL of anisomycin (lines 156-168). It is difficult to believe that such a volume would remain restricted to the CA1 hippocampal subregion, even if injection cannula tips were accurately placed. Without data confirming anisomycin spread, conclusions regarding the specificity of anisomycin effects within the CA1 region are questionable. Please provide data or justification regarding drug diffusion within the CA1 region.

Answer 1: In this study, 20 μg/μl of anisomycin (ANI) was injected into the bilateral hippocampal CA1 regions at a flow rate of 0.5 μl/min for 2 minutes, resulting in a total volume of 1 μl per side. In short, 20 μg of ANI was administrated into each side of the CA1. A previous study reported that the same amount of ANI injected into this region inhibits protein synthesis not only in the CA1 region but also in adjacent regions such as the DG, although this suppression is confined to the dorsal hippocampus (Wanisch, 2008). Based on this, rephrased the descripion “protein synthesis in the hippocampal CA1 region” to “protein synthesis in the dorsal hippocampus” throughout the paper.

Comment 2: [2] Exploration Time per Object: The absolute exploration time per object alone is insufficient to accurately reflect cognitive ability, as it can be confounded by individual differences in general activity, motivation, or anxiety. Relative measures such as discrimination ratios would better represent true cognitive performance. Please address this limitation clearly.

Answer 2: As the reviewer pointed out, “absolute exploration time per object” does not accurately reflect memory ability due to individual differences between rats. Therefore, to eliminate the influence of individual differences, we calculated the discrimination rate (DR) and presented those results. Additionally, as the previous version of the manuscript did not clearly state that DR is considered a more reliable indicator of memory performance than absolute exploration time adjusted for individual variability, In the revised manuscript, the Discussion section has been revised to emphasize the importance of DR (ll. 571-575).

Comment 3: [3] Statistical Analyses [3.1] Number of Animals and Repeated Testing: Experimental designs depicted in Figure 1 and its legend are unclear regarding repeated testing. Were all animals tested multiple times?

• Experimental designs depicted in Figure 1 and its legend are unclear regarding repeated testing. Were all animals tested multiple times?

• For Experiments 1 and 2, the number of samples shown in Figures 5 and 6 (19 and 15 samples, respectively) does not align with the reported total of 40 and 30 animals (line 133). Clarify whether animals underwent two or four tests. Additionally, repeated measures ANOVA is only appropriate if all animals experienced every condition; otherwise, a mixed-model analysis would be required. Please clarify these discrepancies and justify your choice of statistical method.

Answer 3: We apologize for any confusion caused by the lack of explanation about experimental designs and statistical analyses. First, we revised the figures showing the experimental procedures so that it is clear how the OL test was repeated for each rat in each experiment. Briefly, all rats underwent the OL test under each condition within their respective experiments. The relevant figures have been revised and are also now displayed together with the results for each experiment.

In addition, this study used 20, 20, 40, and 30 rats for Experiments 1 to 4, respectively. Of these, rats with poor activity during the OL test (1 in Expt. 1 and 2 each in Expt. 2 & 3), a blocked catheter (2 in Expt. 1), poor running during exercise (1 in Expt. 4), or failed cannula placement surgery (1 in Expt. 3) were excluded from analyses (ll. 299-303). The final sample sizes (N) for each experiment are clearly stated in the figure legends.

Additionally, there was an incorrect description of the statistical analysis procedure which has been corrected: As the reviewer pointed out, a repeated measures ANOVA should be applied when the same rats experience all conditions, and a mixed-model ANOVA should be applied when there are factors without repetition (see the Statistical Analysis section, ll. 303-318).

Comment 4: [3.2] Information in Figures and Legends: Clearly indicate the timing and number of repeated tests in Figure 1 and legends of all relevant figures. Include explicit sample sizes (N numbers) for each group in all figures or legends.

Answer 4: In the previous manuscript, all experimental procedures were collectively presented in Figure 1; however, in the revised manuscript, they are shown separately in the respective experimental figures (Fig 1A, and 3-6A). The procedures were revised to clarify the timing and number of repeated OL tests. Additionally, final sample sizes (N) for each group have been added to the figure legends.

Comment 5: [3.3] Validation of Behavioral Paradigm: Given the high variability observed in total distance traveled and object exploration, analyze whether repeated exposure to the task affected behavior (e.g., reduced exploration due to familiarity or increased spatial understanding). If these analyses were performed previously, reference them clearly. Clarify protocols regarding repetition: were objects, arenas, or spatial cues changed? Was the interval between sessions consistently two days?

Answer 5: To confirm the effects of repeated exposure to the OL task on exploration behavior, we compared locomotor activity (total distance moved) and exploratory behavior (total exploration time) between the 1st and 2nd OL test trials in each experiment. Although the outcomes were not entirely consistent across experiments, the 2nd OL trial tended to show a decrease in exploratory behavior. This suggests that repeated exposure to the OL task may induce environmental adaptation, leading to reduced spontaneous exploratory behavior—a phenomenon also observed in other behavioral tasks such as the spontaneous recognition test (Broadbent et al., 2010) and the open field test (Chen et al., 2023). Importantly, this study assessed memory persistance using DR, a measure that is relatively unaffected by differences in exploration levels (Akkerman et al., 2012). Therefore, the overall interpretation of the results remains valid. These results and related discussion have been incorporated into the revised manuscript (ll. 604-615).

In addition, we have clarified the differences between the 1st and 2nd OL trials in the Methods section (ll. 247-250), and the interval between them in each experimental figure (Fig 3-6A). Briefly, in the 2nd OL trial, the arena and spatial cue (a sidewall with a black-and-white striped pattern) were kept consistent with the first trial, while the two identical objects were replaced with new ones. The interval between trials was 48–52 hours in Experiments 2–4, and 48–70 hours in Experiment 1a.

Comment 6: [3.4] Figure 4D, Lines 531-534: Since no significant difference between exercise and control groups was found, despite exercise group performance being significantly above chance, it remains unclear whether moderate-intensity exercise truly enhances memory retention. Adjust interpretations accordingly.

Answer 6: We also consider it a question that, despite the DR in the AME intervention group exceeding the chance level in Expt. 2 and 4, no difference was observed between the AME group and other groups. The reasons for the lack of differences between groups remain unclear. However, with spontaneous object exploration tasks, it has been suggested that it may be more appropria

---

## [Editor Report · Decision Letter 1]

Memory persistence enhancement by post-learning moderate exercise requires de novo protein synthesis in the dorsal hippocampus

PONE-D-25-16208R1

Dear Dr. Inoue,

We’re pleased to inform you that your manuscript has been judged scientifically suitable for publication and will be formally accepted for publication once it meets all outstanding technical requirements.

Kind regards,

Etsuro Ito, Ph.D.

Academic Editor

PLOS ONE

---

## [Editor Report · Acceptance letter]

PONE-D-25-16208R1

PLOS ONE

Dear Dr. Inoue,

I'm pleased to inform you that your manuscript has been deemed suitable for publication in PLOS ONE. Congratulations! Your manuscript is now being handed over to our production team.

Kind regards,

on behalf of

Prof. Etsuro Ito

Academic Editor

PLOS ONE